# IFNα Subtypes in HIV Infection and Immunity

**DOI:** 10.3390/v16030364

**Published:** 2024-02-27

**Authors:** Zehra Karakoese, Martha Ingola, Barbara Sitek, Ulf Dittmer, Kathrin Sutter

**Affiliations:** 1Institute for Virology, University Hospital Essen, University of Duisburg-Essen, 45147 Essen, Germany; zehra.karakoese@uk-essen.de (Z.K.); ulf.dittmer@uk-essen.de (U.D.); 2Institute for the Research on HIV and AIDS-Associated Diseases, University Hospital Essen, University of Duisburg-Essen, 45147 Essen, Germany; 3Medical Proteome Center, Ruhr University Bochum, 44801 Bochum, Germany; martha.ingola@ruhr-uni-bochum.de (M.I.); barbara.sitek@rub.de (B.S.); 4Department of Anesthesia, Intensive Care Medicine and Pain Therapy, University Hospital Knappschaftskrankenhaus Bochum, 44892 Bochum, Germany

**Keywords:** HIV, type I IFNs, IFNα subtypes, immunotherapy

## Abstract

Type I interferons (IFN), immediately triggered following most viral infections, play a pivotal role in direct antiviral immunity and act as a bridge between innate and adaptive immune responses. However, numerous viruses have evolved evasion strategies against IFN responses, prompting the exploration of therapeutic alternatives for viral infections. Within the type I IFN family, 12 IFNα subtypes exist, all binding to the same receptor but displaying significant variations in their biological activities. Currently, clinical treatments for chronic virus infections predominantly rely on a single IFNα subtype (IFNα2a/b). However, the efficacy of this therapeutic treatment is relatively limited, particularly in the context of Human Immunodeficiency Virus (HIV) infection. Recent investigations have delved into alternative IFNα subtypes, identifying certain subtypes as highly potent, and their antiviral and immunomodulatory properties have been extensively characterized. This review consolidates recent findings on the roles of individual IFNα subtypes during HIV and Simian Immunodeficiency Virus (SIV) infections. It encompasses their induction in the context of HIV/SIV infection, their antiretroviral activity, and the diverse regulation of the immune response against HIV by distinct IFNα subtypes. These insights may pave the way for innovative strategies in HIV cure or functional cure studies.

## 1. Introduction

Type I interferons (IFN) belong to a pleiotropic cytokine family and are rapidly induced by viral infections. They bind to their ubiquitously expressed IFNα/β receptor (IFNAR), consisting of the two subunits, IFNAR1 and IFNAR2. This binding activates the classical Jak (Janus kinases)-STAT (signal transducers and activators of transcription proteins) signaling cascade, which leads to the transcription of hundreds of IFN-stimulated genes (ISGs). During infection with certain viruses, specific patterns of ISGs are expressed, resulting in distinct antiviral activities for each virus [1]. These activities include the expression of directly acting ISGs, so-called viral restriction factors, as well as the repression of cellular dependency factors, so-called IFN-repressed genes (IRepGs) [2,3,4]. In addition to these more direct antiviral effects, type I IFNs also modulate virus-specific innate and adaptive immune responses by promoting the differentiation and activation of innate and adaptive immune cells.

Type I IFNs belong to a multigene family consisting of several IFNα subtypes but only one IFNβ, IFNε, IFNκ, and IFNω (human), or limitin (mouse) [5]. IFNα subtype genes exist in all vertebrates [6,7], and they likely developed from an ancestor *IFNA*-like gene by gene conversion and duplication [6,7]. All 13 human *IFNA* subtype genes (*IFNA1*, *IFNA2*, *IFNA4*, *IFNA5*, *IFNA6*, *IFNA7*, *IFNA8*, *IFNA10*, *IFNA13*, *IFNA14*, *IFNA16*, *IFNA17*, and *IFNA21*) are located on chromosome 9 [8,9,10] and encode for 12 different IFNα subtype proteins, with identical sequences of mature IFNα1 and IFNα13, thus referred to here as IFNα1. The human IFNα subtypes have similarities in structure: they lack introns, they have similar protein lengths (165–166 amino acids), and their protein sequences are highly conserved (75–99% amino acid sequence identity) [11,12]. The IFNα subtypes all bind to the same IFNα/β receptor, but they differ in their binding affinity to both receptor subunits [13]. This may be associated with differences in downstream signaling events, including the phosphorylation of distinct STAT molecules and mitogen-activated protein kinases (MAPK), which were reported after stimulation of cells with individual subtypes [14,15]. Furthermore, there is growing evidence that cell type specificities, the microenvironment, receptor avidity, timing, and fine-tuning of downstream signaling events, all contribute to the complex biology of IFNα subtypes [16,17]. This ultimately results in distinct antiviral and immunomodulatory properties of individual subtypes in different viral infections [18,19,20,21,22,23,24,25,26]. Here, we summarize the growing body of literature on the biological role of IFNα subtypes in retroviral infections, with a special focus on HIV and SIV infections. Specifically, in this review, we discuss the induction of IFNα subtype expression by retroviruses, their antiretroviral capacity, and their impact on innate and adaptive immune responses against retroviruses.

## 2. Induction of IFNα Subtypes during Retroviral Infections

Throughout the HIV life cycle, diverse replication intermediates, including ssRNA, dsRNA, DNA:RNA hybrids, and dsDNA, coexist. Distinct pattern recognition receptors (PRRs), such as Toll-like receptor (TLR) 7/8 [27], cyclic GMP-AMP synthase (cGAS) [28], DEAD-box polypeptide 3 (DDX3) [29], and retinoic acid inducible gene I (RIG-I) [30], can sense these replication intermediates of HIV. After the binding of various nucleic acid ligands to their respective receptors, different signaling cascades are initiated, which require the binding of adaptor molecules such as MyD88 (Myeloid differentiation primary response gene), MAVS (Mitochondrial antiviral-signaling protein), or STING (Stimulator of Interferon Genes) to their receptors. This further activates different kinases like IRAKs (interleukin-1 receptor-associated kinases) or TBK1 (TANK-binding kinase 1), ultimately resulting in downstream phosphorylation of the transcription factors IFN regulatory factor (IRF) 3 and 7, which trigger the production of type I IFNs (Figure 1). The promoter regions of *IFNA* genes contain positive regulatory domains (PRD) I and PRD III-like elements [11], serving as binding sites for IRF family members. During early infection, *IFNB* and *IFNA1* (human) or *Ifna4* (mouse) are exclusively expressed through IRF3, triggering the expression of IRF7, which is required for the transcription of the other *IFNA* subtypes. During viral infection, IRF7 expression increases, while plasmacytoid dendritic cells (pDCs), which are characterized by high basal IRF7 expression, trigger rapid transcription of *IFNA* genes [31,32]. IRF3 and IRF7 bind to different IRF elements within the viral response element of the *IFNA* gene promoters. These different IRF elements can be targeted by both IRF3 and IRF7 or are selective targets for one of these factors. Thus, IRF3/7 binding to the different signaling elements regulates differential *IFNA* gene expression, which depends on the relative expression and ratio of IRF3 and IRF7 in different cell types. These factors can change in cell types during an ongoing virus infection [9,33].

During acute HIV infection, a rapid and transient IFN response was measured systemically, peaking between days 5 and 15 post HIV infection [34]. Interestingly, the induction of IFNs is positively correlated with plasma viral loads [34]. Previous studies analyzed *IFNA* subtype gene expression in pDCs or peripheral blood mononuclear cells (PBMCs) from HIV-infected individuals at the mRNA level [35,36] (Table 1). A comparison between healthy individuals and patients with HIV (CDC stage A or C) revealed increased expression of *IFNA6* and *IFNA2* mRNA in patients with HIV. Additionally, patients with Acquired Immunodeficiency Syndrome (AIDS; stage C) showed significantly higher expression of *IFNA1/13*, *IFNA8*, *IFNA14*, *IFNA16*, *IFNA17*, and *IFNA21* mRNA. *IFNA2* mRNA was strongly elevated during HIV infection and inversely correlated with CD4^+^ T cell counts [35]. Similar results were found in a study comparing gene expression profiles of *IFNA* subtypes in chronically HIV-positive individuals under ART or treatment-naïve individuals [36]. Seven *IFNA* subtypes (*IFNA2*, *IFNA4*, *IFNA5*, *IFNA6*, *IFNA7*, *IFNA14*, and *IFNA16*) accounted for over 95% of the total IFNα mRNA response in all chronically HIV-positive individuals, with remarkably similar expression patterns between individual patients, suggesting a common signature of *IFNA* subtypes in humans. Harper et al. analyzed the *IFNA* subtype expression in isolated pDCs exposed to HIV-1_BaL_ using mRNA sequencing [37]. They found a predominant expression of five *IFNA* subtype mRNAs in exposed pDCs (*IFNA1/13*, *IFNA2*, *IFNA5*, *IFNA8*, and *IFNA14*), with three of those being already expressed in mock-stimulated pDCs (*IFNA1/13*, *IFNA5*, *IFNA8*) [37]. In addition, increased levels of *IFNA1/13*, *IFNA2*, *IFNA5*, and *IFNA4* were detected in PBMCs from untreated, chronically HIV-1-positive individuals in comparison to uninfected controls. In this study, *IFNB* mRNA was not detected in PBMCs from infected patients [38]. Interestingly, in gut-derived lamina propria mononuclear cells (LPMCs) from the same patients, a decrease in *IFNA* transcripts (all individual subtypes) was observed, whereas *IFNB* transcripts were elevated in patients with HIV [38]. Comparing these different studies revealed both shared (*IFNA1/13*, *IFNA2*, *IFNA5*) and distinctly HIV-induced subtypes (*IFNA4*, *IFNA6*, *IFNA7*, *IFNA8*, *IFNA14*, *IFNA16*). These findings suggest that the cell type, the infecting virus isolate, and the sensing pathway may collectively regulate the expression of individual IFNα subtypes during HIV infection.

Some studies also explored the induction of *IFNA* genes during acute SIV infection in rhesus or pigtailed macaques [39,40,41]. Although simian *IFNA* genes cannot be directly compared with human genes, these studies revealed some interesting insights. In rhesus macaques, tissue-specific expression of *IFNA* subtype transcripts was observed in various organs [39]. In the thymus, the expression pattern changed rapidly during acute SIV infection, with certain subtypes consistently expressed. In orally SIV-infected infant rhesus macaques, despite similar high viral loads, *IFNA* transcription differed between lymphoid and mucosal tissues, with a strong induction in lymphoid tissues and only slight increases in mucosal tissues [40]. In a pigtailed macaque model with neurological manifestations after SIV infection, a positive correlation between *IFNA* gene expression and viral loads was demonstrated in organs with high viral replication. High *IFNA* expression was attributed to pDC infiltration in these organs [41].

All reports on HIV/SIV infection showed different results in *IFNA* subtype gene expression, dependent on the analyzed tissue, cell type, or stimulus. Despite the importance of tissue or cell specificity in studies on IFNα subtypes, only minor differences between individual test samples from patients with HIV were found in one study. This suggests that genetic diversity between individuals has a rather low impact on the *IFNA* subtype expression pattern after virus infection. 

It is noteworthy that so far all studies rely on mRNA expression levels for subtype differentiation, as protein-based analysis (e.g., ELISA, ultrasensitive single-molecule array (Simoa^®^) [42], Western Blot) is currently not feasible due to the lack of IFNα subtype-specific antibodies. Therefore, a novel assay to distinguish and quantify the IFNs at the protein level (e.g., using mass spectrometry (MS) analysis) is required in the future and should be developed.

## 3. IFNα Subtype-Mediated Downstream Signaling and ISG Expression Pattern during HIV Infection

The expressed and secreted IFNα subtypes all bind to a common heterodimeric type I IFN receptor, consisting of the subunits IFNAR1 and IFNAR2. In general, IFNα subtypes have a higher binding affinity to IFNAR2 (KD: 0.4–5 nM; except for IFNα1—220 nM) than to IFNAR1 (KD 0.5–5 µM) [13], indicating an initial binding to IFNAR2, which then recruits IFNAR1 to form the ternary complex [43,44]. The different subtypes have various binding affinities to both receptor subunits; however, the binding affinities do not necessarily reflect the antiviral activity (tested against VSV or EMCV) of the individual subtypes [13]. The binding affinities to IFNAR2 are comparable for all subtypes, with the exception of IFNα1, with a binding affinity that is more than 130-fold lower compared to IFNα2 [13]. The product of the binding affinities to both receptor subunits (IFNAR1 and IFNAR2) of IFNα2, IFNα4, IFNα5, IFNα10, IFNα17, and IFNα21 are comparable, whereas the subtypes IFNα7, IFNα8, and IFNα16 have a three to four times higher binding affinity, and IFNα6 and IFNα14 have an eight times higher binding affinity compared to IFNα2. The two outliers are IFNα1 and IFNβ, with an affinity that is 40 times lower and 1000 times higher, respectively, than that of IFNα2 [44]. The formation of the ternary complex leads to the activation of the canonical JAK-STAT pathway, although signaling through IFNAR1 alone by IFNβ has recently also been suggested [45]. Since IFNAR lacks intrinsic kinase activity, it relies on the receptor-associated protein JAK1 and tyrosine kinase 2 (TYK2) to phosphorylate STAT1 and STAT2 [46], followed by the heterodimerization of STAT1-STAT2, which recruits IRF9 to form the IFN-stimulated gene factor 3 (ISGF3) complex (Figure 2). ISGF3 then translocates into the nucleus, where it binds to a conserved genomic sequence motif (about 15 bp), called the IFN-stimulated response element (ISRE), located in the promoter region of numerous IFN-stimulated genes [47,48]. IRF9 is a key factor for transcriptional regulation, as it provides the specificity for binding to ISRE, which regulates the transcription of hundreds of ISGs and thereby the establishment of an antiviral state in cells or even whole organs [49]. Recently, the homeostatic chromatin state of ISRE was shown to be cell type-specific, resulting in cell type-specific differences in ISRE binding patterns upon IFN stimulation [50].

The antiviral state in cells is mainly induced by the canonical signaling pathway described above, which results in the expression of many ISGs that contribute to IFN-specific biological activity [51]. However, IFNα can also signal through non-canonical pathways (Figure 2). The activation of these non-canonical pathways may lead to profound differences in ISG expression patterns [52]. STAT1-STAT1 homodimers or other STAT dimers, such as STAT3, and STAT5A, can be activated by IFNα subtypes and are part of STAT-dependent non-canonical pathways. STAT4 and STAT6 appear to be restricted to certain cell types but can also be activated by IFNα [53,54]. These complexes, especially STAT1 homodimers, can bind to the IFNγ-activated site (GAS) element, which is present in the promoter region of certain ISGs [55]. Some ISGs only have ISRE or GAS elements in their promoter region; however, some ISGs have both elements, indicating that the ISG pattern induced by individual IFNα subtypes may vary according to their STAT signaling and promoter element activation [53]. We showed, for example, that IFNα14 induces STAT1:STAT2 heterodimer signaling as well as STAT1:STAT1 homodimer activation of GAS elements [24]. This combined type I and II IFN signaling resulted in the induction of 844 ISGs in hepatoma cells, whereas the only canonical type I IFN signaling by IFNα2 induces only 325 ISGs. This large set of additionally induced ISGs by the IFNα14 subtype was at least correlated with its strong antiviral activity against HIV [56] and HBV [24]. Similar findings were made in HIV target cells, human lamina propria CD4^+^ T cells [57]. Here, IFNα2 induced only 302 ISGs (including 266 core ISGs expressed by all five tested IFNα subtypes in the study), whereas the more antiviral IFNα14 induced a large number of 509 additional ISGs measured by RNA sequencing technology. In addition to the STAT signaling pathways, STAT-independent downstream signaling, such as MAPK and phosphoinositide 3-kinases (PI3K), showed activation upon type I IFN binding to IFNAR [58]. The mammalian target of rapamycin (mTOR) pathway, which mediates mRNA translation, can be activated downstream of the PI3K/AKT pathway [53]. Also downstream of this pathway is p38, which is rapidly activated in response to IFNα, without modifying the activation of the STAT pathway, and has been demonstrated to be crucial for the antiviral function of IFN [59]. Another non-canonical pathway is the extracellular signal-regulated kinase (ERK) signaling cascade. In contrast to p38 downstream signaling, this pathway has not yet been investigated thoroughly [60]. However, cell-specific activation of this pathway upon IFNα treatment has been demonstrated [61]. Interestingly, HIV has been shown to use both pathways (p38 and ERK) to deplete CD4^+^ T cells from the immune system as well as to produce new virions [62], indicating a potential influence of IFN on T cell depletion or viral replication. However, IFNα has been shown to inhibit HIV latency and even reverse established latency in a STAT1-, STAT3-, and/or STAT5-dependent manner, independent of NFκB activation [63].

To analyze changes in signaling pathways, researchers often investigate posttranslational modifications (PTMs), especially phosphorylation, since they are important in signal transduction and many cellular processes (reviewed in [64]). Currently, the characterization and quantification of phosphorylated peptides and proteins are performed using well-established high-throughput MS-based phosphoproteomics, which has proven to be particularly useful for simultaneously monitoring numerous phosphoproteins within different signaling networks [65]. Phosphoproteomics has been used successfully to screen for primary human CD4^+^ T cells after HIV-1 infection, resulting in a global view of the signaling events induced during the first minute of HIV-1 entry [66]. Other ways to analyze the phosphorylation of signaling events in a more targeted manner are using phosphoflow cytometry or western blots. Phosphoflow analysis of T and NK cells revealed strong differences in STAT1 and STAT5 phosphorylation after treatment with IFNα2, IFNα14, and IFNβ. IFNα14 and the high-affinity IFNβ significantly increased the frequencies of phosphorylated STAT1, STAT3, and STAT5 in the gut- and blood-derived T and NK cells, whereas a higher activation of pSTAT5 was observed in PBMCs and a higher STAT1 phosphorylation in LPMCs. Additionally, significant differences in the phosphorylation of STAT5 were observed in both healthy and HIV-infected individuals, indicating an IFNα subtype-specific potency to stimulate T and NK cell responses during HIV-1 [67]. In addition, western blot analysis of IFN-stimulated murine CD8^+^ T cells demonstrated strong phosphorylation of STAT1 and STAT2 by murine IFNα6 and IFNα11, which was completely undetectable in CD8^+^ T cells after stimulation with murine IFNα1 and IFNα2 [68]. Furthermore, tyrosine phosphorylation of STAT1 was also induced in response to murine IFNα1, IFNα2, IFNα4, and IFNα5 in J2E erythroid cells, while tyrosine phosphorylation of STAT3 was induced only in response to IFNα1. This indicates cell-type-specific differences in the activation of different STAT molecules by various IFNα subtypes. 

All of the above demonstrates the complexity of the downstream signaling of IFNα and suggests that there may still be undefined mechanisms that mediate cellular IFN responses. Also, a more detailed understanding of how infections such as SIV/HIV utilize these pathways to their benefit is needed, which may provide insights into their pathogenesis. Finally, further characterization of each IFNα subtype and activation of downstream signaling cascades is needed, since this may provide important insight on their antiviral and immunomodulatory diversity. 

## 4. Antiviral Activity of IFNα Subtypes during Retroviral Infections

The orchestration of an effective immune response during infection hinges on the induction of a multitude of ISGs [52]. These genes have direct or indirect antiviral activities and are crucial for suppressing various stages of the viral life cycle, including entry, reverse transcription, translation, packaging, and release of newly synthesized virions. However, the intricacies of target specificity and the mechanisms of action for many ISGs remain enigmatic. A seminal study conducted twenty-five years ago marked the origin of understanding the anti-HIV-1 activity exhibited by different recombinant IFNα subtypes and artificial IFN-derived mutants [69]. In this investigation, the 50% inhibitory concentration (IC50) against HIV replication in MT-2 cells exhibited a broad spectrum, ranging from remarkably low concentrations for an IFN hybrid derived from IFNα7 and IFNα10, as well as a point mutation in IFNα7 at Ser116, up to approximately 6000 times higher concentrations for IFNα1 [69]. Subsequent research delved into the antiviral effects of all human IFNα subtypes against HIV-1_NL4-3_ on PBMCs in vitro. Notably, certain IFNα subtypes, such as IFNα14, IFNα6, IFNα17, and IFNα21, exhibited potent inhibition of viral replication, as evidenced by reduced cellular p24 levels and infectivity of cell culture supernatants [56]. Interestingly, the clinically improved subtype IFNα2a/b, commonly used in hepatitis B virus (HBV) therapy, only modestly suppressed HIV replication in vitro, which was in line with previous clinical trials using IFNα2a/b against HIV in infected patients [70,71,72,73]. A study by Tauzin and colleagues also investigated the individual inhibition of HIV by different IFNα subtypes, focusing on distinct steps of the viral replication cycle. Their study revealed that although all subtypes exhibited similar abilities to block virus entry, they differed in their effectiveness in inhibiting other early stages of HIV replication. IFNα10, IFNα14, IFNα16, and IFNα17 were very potent in restricting DNA synthesis, while IFNα1, IFNα2, and IFNα21 were the least effective in this regard. Furthermore, only individual subtypes demonstrated efficient targeting of the later stages of replication, like viral protein synthesis (notably IFNα6, IFNα8, IFNα16, and IFNα17 followed by IFNα2, IFNα5, and IFNα14) or virus release (IFNα14 and IFNα21). The study suggests that IFN treatment might be less effective in inhibiting the late stages of the HIV replication cycle compared to the early stages. Additionally, only a specific subset of IFNα subtypes demonstrates significant viral inhibition [74], confirming their qualitative differences in anti-HIV activity [56]. Expanding these investigations with LPMCs further confirmed the high antiviral potency of specific IFNα subtypes [37]. IFNα8, IFNα14, IFNα6, IFNα17, and IFNα10 were identified as the most effective inhibitors of HIV-1_BaL_ replication in gut-derived LPMCs. Again, stimulation with IFNα2, a subtype that is prominently induced during HIV infection in patients, resulted only in a partial reduction of viral replication in LPMCs. Utilizing the mucosal LPMC HIV-1 infection model revealed a correlation between the antiviral potency of IFNα subtypes and their pattern of induced ISGs. Particularly, potent antiviral IFNα subtypes, such as IFNα8 and IFNα14, induced high mRNA expression of *Mx2* and *Tetherin* [37], two well-known HIV restriction factors. In contrast, weak antiviral subtypes, like IFNα1 and IFNα2, induced low or no enhancement of mRNA expression for these ISGs. Interestingly, both weak (IFNα1) and potent IFNα subtypes (IFNα8) enhanced APOBEC3G-mediated hypermutations against HIV-1 in LPMCs [37]. This was in contrast to the results of humanized mice, where an increased hypermutation rate was detected in lymph nodes after treatment with the strong antiviral IFNα14, but not with the weak antiviral IFNα2 [56]. Furthermore, the interferome analysis of primary gut CD4^+^ T cells, a prime target for HIV, stimulated ex vivo with individual human IFNα subtypes and IFNβ revealed only a low number of core ISGs that were induced by all tested type I IFNs, whereas the subtype with the highest number of additional ISGs was IFNα14 [57], nicely correlating with its high anti-HIV potency. In an in vitro latency model, pDCs hindered HIV latency establishment via secretion of type I IFNs (IFNα, IFNβ, IFNω). However, once latency was established, only IFNα, no other type I IFNs, efficiently reversed latency in both the in vitro model and CD4^+^ T cells from people living with HIV (PLWH) on suppressive ART, indicating diverse roles of type I IFNs at different stages of HIV infection [63].

The role of the C-helix region of the IFN molecule for antiviral activity was elucidated through in vitro stimulation of HIV-1_BaL_-infected macrophages. IFNα2, IFNα21, and hybrids or mutants derived from these subtypes demonstrated that the C-helix region (Figure 3; dark purple cylinder), specifically amino acids 81–95, may play a crucial role in antiviral activity against HIV. Notably, a hybrid (Hy1) IFN molecule composed of the N-terminus of IFNα21 and the C-terminus of IFNα2 exhibited the highest potency in inhibiting HIV replication [75]. Comparing the amino acid sequences of the potently antiviral IFNα subtypes (IFNα6, IFNα14) with the clinically approved IFNα2b, critical differences in the binding sites to IFNAR1 and IFNAR2 were identified [15]. IFNα6 had only minor differences in the IFNAR1/2 binding sites, while IFNα14 showed multiple differences in IFNAR1/2 binding sites and a putative ‘tunable anchor’ region (located in helix B; Figure 3; pink cylinder). This putative ‘tunable anchor’ region is characterized by a conserved binding site with IFNAR1, but a more variable region at the side facing the core of IFNα, which might modulate the fine structure of the IFN by mutations in the core. The amino acid exchange between IFNα14 and IFNα2 improved the antiviral activity of hybrid IFN against HIV in vitro [15]. Helices B-D (Figure 3; pink cylinder), especially IFNAR1 binding sites and the ‘tunable anchor’, were shown to be crucial for improving the antiviral activity. Similar effects were observed with another IFNα2-mutant, with the mutations H57Y, E58N, and Q61S in helix B (Figure 3; pink, dark purple, and mineral green cylinders), which showed a 60-fold increase in IFNAR1 binding affinity and a 3.5-fold increase in antiviral activity against the VSV [76], further underlining the importance of binding motifs to IFNAR1 for the antiviral activity of IFN.

Transitioning from in vitro studies to in vivo investigations, differences in the antiviral activity of murine IFNα subtypes were observed in acute and chronic Friend retrovirus (FV) infection [18,19,78], suggesting specific antiretroviral activities of distinct subtypes rather than a pan-antiviral activity of all subtypes against retroviruses in vivo. To further scrutinize these subtype-specific antiviral activities against HIV in vivo, their therapeutic potential was explored using C57BL/6 Rag2^-/-^ γc^-/-^ CD47^-/-^ bone marrow-liver-thymus (BLT) humanized mice infected with HIV-1_JR-CSF_ [56]. Treatment with human recombinant IFNα14, identified as the most potent subtype in vitro, significantly decreased HIV-1 replication in vivo, as evidenced by reduced plasma p24 and plasma RNA copies during both acute (11 dpi) and established (45 dpi) HIV-1 infections. In contrast, IFNα2 treatment had no antiviral effect in this mouse model, which was in line with the results of in vitro studies. Interestingly, HIV-1 proviral loads were significantly reduced by both IFNα14 and IFNα2, with IFNα14 being more effective [56]. In another humanized mouse model utilizing NOD-scid IL2rγcnull (NSG) mice implanted with human PBMCs (Hu-PBL mice), animals were infected with HIV-1_BaL_, and sustained expression of IFNα was induced through hydrodynamic injection of plasmids encoding different IFNα subtypes (*IFNA2A*, *IFNA6*, *IFNA8*, *IFNA14*, and *IFNB*). The results indicated that IFNα14 and IFNβ significantly reduced plasma p24 levels, while IFNα2a, IFNα6, and IFNα8 only slightly inhibited HIV replication [79]. Combining oral antiretroviral therapy (cART) with IFNα14 in chronically HIV-1-infected BLT-humanized mice further suppressed HIV-1 plasma viremia in humanized mice compared to cART alone, but failed to notably reduce the proviral DNA reservoir [80]. A comparable study in humanized mice with pasylated IFNα14 showed no effect of single IFNα14 treatment during chronic HIV infection; however, previous cART reduced ISG expression in chronically infected mice, and subsequent IFNα14 therapy resulted in a transiently lower HIV burden [81]. 

The studies indicate that certain IFNα subtypes may be more effective in controlling HIV infection than others. However, some studies demonstrated the detrimental effects of IFNs in HIV infection, emphasizing a knowledge gap regarding the optimal subtype and timing of administration during suppressive cART that results in a beneficial versus detrimental outcome of IFN treatment in HIV infection. Sandler et al. observed desensitization and an increased viral reservoir size in SIV-infected macaques after continuous IFNα2a treatment during the acute infection phase [82]. This suggests that too much IFNα during acute retroviral infection might be detrimental. On the other hand, endogenous IFNα responses are obviously important, as the application of an IFNAR antagonist that blocks type I IFN responses in acutely SIV-infected macaques led to reduced antiviral gene expression, increased SIV reservoir size, and accelerated CD4 T cell depletion [82]. A study utilizing an IFNα blocking antibody (AGS-009), which blocks 11 out of 13 rhesus macaques IFNα subtypes, during acute SIV infection of rhesus macaques reported a modest increase in viral replication and a trend toward faster development of AIDS. This underscores the significance of IFNα activity during acute SIV infection [83].

However, studies during persistent HIV or SIV infection also led to some contradictory results. Treatment of chronically SIV-infected sooty mangabeys with recombinant rhesus macaques IFNα2 results in an up to 10-fold decrease in SIV viremia and a strong ISG induction early during treatment [84]. However, during sustained therapy (up to 4 months of IFN administration), viremia increased and ISG expression decreased again, suggesting a state of tolerance against exogenous IFNα induced after weeks of treatment [84]. This might also be the case for sustained endogenous type I IFN responses during chronic retroviral infection. The following studies showed that IFNAR blocking studies in humanized mice using anti-IFNAR1 or anti-IFNAR2 antibodies resulted in reduced HIV reservoir size and delayed viral rebound post-cART cessation [85,86]. Along these lines, IFNAR blockade in chronic SIV infection during cART effectively dampened the inflammatory pathways associated with type I IFNs in macaques. Notably, in contrast to the mouse studies, IFNAR blockade did not lead to a compromised ability to control SIV replication [87]. Similar observations were also reported from chronic infections with Lymphocytic Choriomeningitis Virus (LCMV) in mice, in which blockade of IFNAR1 resulted in the control of persistent LCMV infection [88,89]. In a follow-up study by the same authors, they uncovered that blocking of IFNβ is required to control persistent LCMV infection, whereas antibody blockade of IFNα (Clone TIF-3C5, which recognizes multiple murine IFNα subtypes) had no effect on virus control [90]. Only blocking of IFNβ improved T cell responses, decreased the number of infected CD8α-DC, and protected mice from disruption of splenic architecture, suggesting a critical role of IFNβ but not IFNα in the immunopathology of chronic viral infections [90]. Therefore, studies in chronic SIV or HIV infection that demonstrated control of retroviral infection by blocking type I IFN signaling (α-IFNAR) must be viewed with caution, as they do not distinguish between the role of IFNα and IFNβ in persistent HIV/SIV infections. A study specifically targeting rhesus macaque IFNα, rather than the entire type I IFN response in chronically ART-treated SIV-infected rhesus macaques reported that IFNα blockade led to the activation of immune pathways that reduced viral persistence during ART [91]. 

Collectively, these findings suggest that the impact of type I IFN treatment on HIV infection depends on the timing of administration, the individual type I IFN subtype, and the level of inflammation in the infection environment. The antiviral potency of IFNs is, at least in part, attributed to their ability to induce HIV restriction factors. IFNα14, in particular, emerges as a promising candidate for the suppression of HIV in immunotherapy studies, highlighting the potential for tailored therapeutic interventions based on the unique properties of specific IFNα subtypes.

## 5. Modulation of Immune Cell Functions by IFNα Subtypes during Retroviral Infections

The role of IFNα subtypes during HIV infection is much broader than their ISG-mediated direct antiviral activity. Importantly, the induction of IFNα during HIV infection can additionally modulate innate and adaptive immune responses. Type I IFNs can enhance antigen presentation by upregulation of Major Histocompatibility Complex (MHC)-I and MHC-II on antigen-presenting cells, support the activation and differentiation of DCs, and activate NK cells and improve their cytotoxicity [54,92,93]. In adaptive immunity, IFNα has a pivotal role in shaping T helper cell (T_H_1) responses and contributes to the activation and clonal expansion of cytotoxic CD8^+^ T cells [94,95]. However, the characteristics of IFNα-mediated immune responses during acute and chronic HIV infection are controversial (reviewed in [96]). In this context, two opposing aspects have to be discussed: IFN-stimulated antiviral immunity versus hyperimmune activation, a state of immune dysfunction. During an acute HIV infection, type I IFN-mediated immune responses are crucial to control initial viral replication. Experimentally delayed type I IFN responses in acutely SIV-infected rhesus macaques by in vivo blockade of IFNAR resulted in accelerated loss of circulating CD4^+^ T cells, a significant decrease in CD4/CD8 T cell ratio, and reduced frequencies of CCR5^+^ memory CD4^+^ T cells [82]. Furthermore, injection of recombinant rhesus macaque IFNα into African Green Monkeys during acute SIV infection did not induce signs of chronic immune hyperactivation, indicating, that chronic immune activation in SIV infection might be independent of IFN [97]. However, during chronic HIV infection, the role of IFNα remains a topic of debate. Type I IFNs have been discussed to induce hyperimmune activation, which is correlated with the dysfunction and exhaustion of immune cells as well as the loss of CD4^+^ T cells (reviewed in [96]). Interestingly, in colon biopsies of PLWH, an increased gene expression profile for *IFNB* but a decreased gene expression profile for *IFNA* was previously described [38]. The expression of *IFNB* in the gut correlated with gene markers for immune activation and inflammation (*CD38*, *PSMB9*, *NLRC5*, *TNFA*, and *IFNG*), as well as exhaustion (*LAG3*). Furthermore, in the same study, a positive correlation between the expression of ISGs and *IFNB* as well as plasma lipopolysaccharide levels was demonstrated, an indicator for microbial translocation, which in turn is associated with hyperimmune activation [57]. Previously, the depletion of CD4^+^ T cells was shown to be triggered by type I IFN-mediated expression of ISGs and an increase in TRAIL- and Fas-dependent apoptosis [98,99,100]. Therapeutic interventions in PLWH utilizing IFNα2 as monotherapy or in combination with ART yielded inconsistent findings in relation to CD4^+^ T cell counts and viral loads (reviewed in [101]). To date, the majority of research concerning the immunomodulatory properties of IFNα during HIV infection has focused on virus-induced type I IFN or immunotherapies utilizing pegylated IFNα2a/b. However, there is a lack of detailed analysis regarding the specific roles of different IFNα subtypes in modulating antiretroviral immune responses or immune activation.

Finding an appropriate model to investigate IFNα-regulated immune responses in HIV infection remains a hurdle due to the difficult situation with animal models. The generation of humanized mice that are susceptible to HIV provides a reasonable opportunity to investigate IFNα subtype-specific immunomodulation. Using the humanized BLT mouse model, we previously demonstrated differences between the IFNα2 and IFNα14 subtypes regarding their immune activation during acute HIV infection [56]. We noted a significant reduction in plasma CXCL10/IP-10 levels in IFNα14-treated mice compared to untreated controls. CXCL10 is a crucial chemokine associated with HIV-1-induced immune dysregulation [102]. In contrast, the CXCL10 levels in IFNα2-treated mice were comparable to those of untreated controls, suggesting a beneficial impact of IFNα14 on immune activation. Furthermore, neither IFNα2 nor IFNα14 elevated the activation status of CD4^+^ T cells. HIV-induced CD4^+^ T cell depletion, which is associated with hyperimmune activation, was previously investigated in the huPBL mouse model [79]. Humanized mice, infected with HIV-1, were subjected to treatment with plasmids encoding for *IFNA2A*, *IFNA6*, *IFNA8*, *IFNA14*, or *IFNB*. During acute HIV infection (10 days post-infection), all mice treated with IFNs showed no signs of CD4^+^ T cell depletion, while untreated HIV-infected controls exhibited complete CD4^+^ T cell depletion. Interestingly, during chronic HIV infection (40 days post-infection), the therapeutic administration of *IFNA2A*- or *IFNA8*-encoding plasmids was ineffective in preventing CD4^+^ T cell depletion. In contrast, treatment with plasmids encoding for *IFNA6* or *IFNA14* successfully preserved CD4^+^ T cell numbers in HIV-infected mice [79]. In another BLT mouse model for chronic HIV infection, neither IFNα2 nor IFNα14 contributed to the loss of CD4^+^ T cells, and the CD4^+^/CD8^+^ T cell ratio was lower in untreated HIV-infected mice [103]. Thus, these studies with humanized mice show a reduction in HIV-induced hyperimmune activation rather than an acceleration by IFN therapy if the right IFNα subtype is used for treatment. 

Another important feature of type I IFNs is the stimulation of antiviral immune cells. In our HIV infection experiment in BLT humanized mice, we found that the frequencies of cytotoxic CD8^+^ T cells were increased after treatment with IFNα2, while IFNα14 treatment enhanced the frequencies of TRAIL-expressing NK cells [56]. In another study with chronic HIV infection in humanized mice, treatment with IFNα14 led to a reduced activation of CD4^+^ T cells, whereas CD4^+^ T cell activation was increased after IFNα2 treatment. Moreover, a decrease in the expression of markers associated with CD8^+^ T cell dysfunction could be observed after IFNα14 treatment. In contrast, IFNα2 treatment did not reduce the expression levels of T cell exhaustion markers, suggesting that exogenous administration of IFNα2 may not effectively counteract CD8^+^ T cell exhaustion in HIV infection [103]. Both studies conducted in humanized mice revealed cell-specific differences mediated by distinct subtypes of IFNα, with IFNα14 shown to reduce T cell exhaustion and prevent hyperimmune activation. By investigating the role of the different type I IFNs, such as IFNα2, IFNα14, and IFNβ, in in vitro HIV-infected PBMCs and LPMCs, we observed a strong immunomodulatory role of IFNα14 and IFNβ on T cells. In T cells, membrane-bound CD107a is upregulated upon the degranulation of granules containing cytotoxic molecules such as granzymes and perforin and serves as a surrogate marker for cytotoxic activity, which is dysregulated in PLWH [104,105]. Higher frequencies of CD107a-expressing CD4^+^ T cells in LPMCs as well as CD4^+^ and CD8^+^ T cells in PBMCs were detected, while IFNα2 had no effect on degranulating T cells [67]. Additionally, IFNα14 and IFNβ significantly increased TRAIL^+^ CD4^+^ T cell numbers in PBMCs. So far, data from different studies provide rather beneficial effects of IFNα treatment on antiviral immunity during HIV infection without showing any signs of increased hyperimmune activation upon IFN stimulation. However, there are remarkable differences between different IFNα subtypes, emphasizing the need for detailed analyses of each IFNα subtype in HIV infection and therapy. 

## 6. Concluding Remarks

When type I IFNs were discovered, many scientists believed that they represented the golden bullet against many infections. However, 67 years later, many features, especially of IFNα subtypes, are still unknown. One problem was that a lot of data on the clinically approved IFNα2 was generated, whereas the other subtypes were largely ignored. Also, the role of IFNα subtypes in HIV infection has only recently been studied. We discuss here that the induction of individual IFNα subtypes during retroviral infection is a very complex process, most likely influenced by many parameters, including the infected cell type, the infecting virus strain, and pathways of viral sensing. More research is needed to better define this multiparameter process because it is very important for intrinsic as well as innate immunity against HIV. Also very relevant for these initial arms of HIV immunity are the signaling pathways that individual IFNα subtypes induce in HIV target cells. Preliminary research on this topic clearly shows that there is much more than the canonical STAT1:STAT2 signaling pathway. Several other signaling pathways are involved, depending on the specific IFNα subtype used for the stimulation of a cell. The different signaling pathways shape the pattern of ISGs that are expressed. Since several of these ISGs are well-known HIV restriction factors or influence innate immunity against HIV, the IFN-induced ISG pattern is most likely crucial to preventing the establishment of HIV infection upon exposure. Thus, it is of utmost importance to understand these IFN-mediated mechanisms because they might provide new tools to prevent HIV infections. After an HIV infection has been established, IFNα subtypes are still very important because they also positively influence the adaptive immune response against HIV, which is very important to provide a time period of virus control. However, during chronic HIV infection, type I IFN responses and IFN treatment have also been associated with hyperimmune activation, T cell dysfunction, inefficient virus control, and CD4^+^ T cell depletion. Recent studies suggest that this might be more associated with IFNβ than IFNα subtypes. However, the therapeutic potential of each IFNα subtype against acute and chronic HIV infection has to be thoroughly tested, and it is not unlikely that some subtypes may have a more beneficial effect, whereas others may have a more detrimental effect. So far, IFNα14 seems to have outstanding potential for anti-HIV activity. Even with this data at hand, one can still question if IFNα subtypes will ever be used for HIV therapy since we have a very potent and effective ART in clinical use. However, ART does not induce HIV cure or functional cure. HIV cure strategies usually aim to develop combination therapies that reactivate the latent virus from the reservoir, stop its replication with ART, and strengthen immunity to then control or eliminate the virus. For such cure strategies, IFNα subtypes might be important, as it has been shown that they can reactivate latent HIV and stimulate potent antiviral immune responses, so they fulfill two of the requirements for HIV cure. These are interesting possibilities for the therapeutic application of IFNα subtypes in HIV infection, but before such applications can be established, more research on IFNα subtypes, which we missed carrying for almost 60 years, is needed.

## Figures and Tables

**Figure 1 viruses-16-00364-f001:**
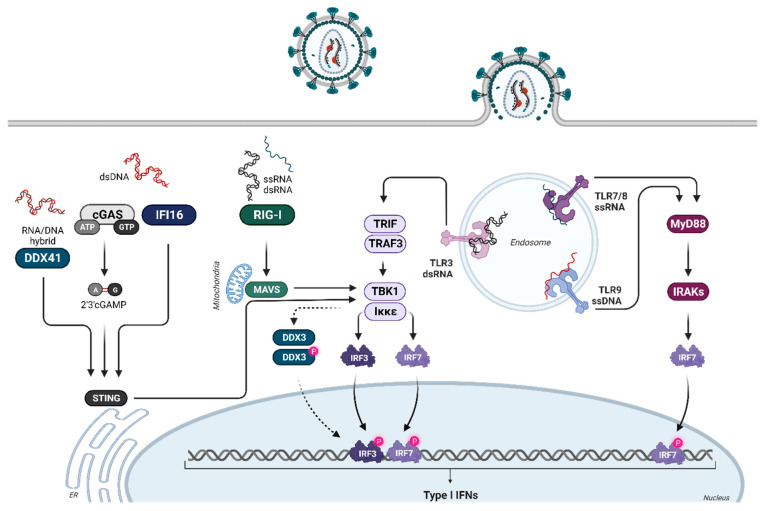
Induction of type I IFNs during retroviral infections. During the HIV life cycle, numerous potential replication intermediates (ssRNA, dsRNA structures, DNA:RNA hybrids, and dsDNA) are present. Some of these are recognized by different PRRs, including TLR7/8, cGAS, DDX3, and RIG-I. Furthermore, the potential sensing of HIV dsRNA structures by TLR3, as well as sensing of HIV DNA by members of the PYHIN family (e.g., absent in melanoma 2 (AIM2) and IFN-γ inducible protein 16 (IFI16)), might also contribute to innate HIV restriction. The detection of HIV DNA or RNA by these different sensors triggers different signaling cascades that lead to the phosphorylation of IRF3 and IRF7. Upon activation, IRF3 and IRF7 translocate to the nucleus, promoting the transcription of type I IFN genes. Created with BioRender.com.

**Figure 2 viruses-16-00364-f002:**
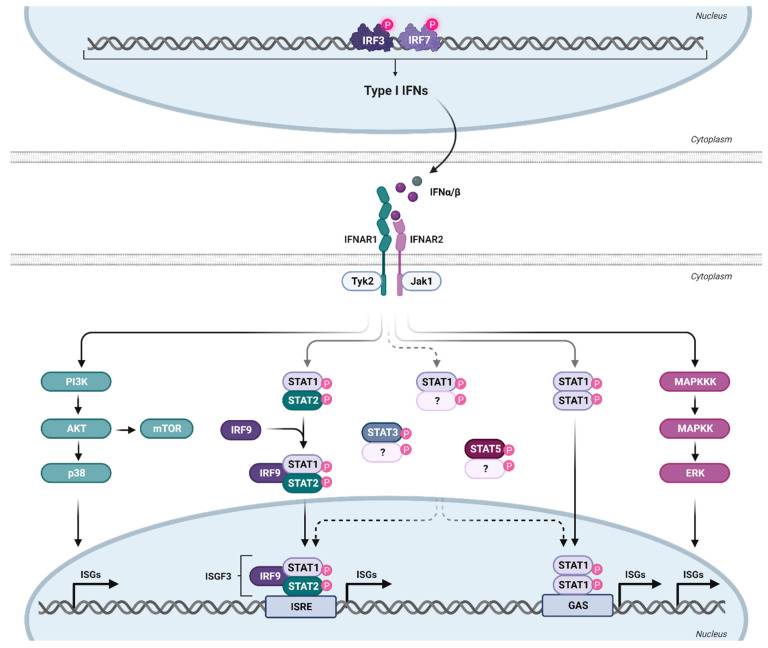
Type I IFN signaling. Binding of type I IFN to the ubiquitously expressed IFNα/β receptor triggers the activation of various signaling cascades. IFNAR consists of the subunits IFNAR1 and IFNAR2, with a higher affinity of IFN for IFNAR2. This leads to initial IFNAR2 binding, followed by IFNAR1 recruitment to form the ternary complex. For canonical signaling, phosphorylation of the receptor unit by Janus kinases (Tyk2 and Jak1) activates transcription factors STAT1 and STAT2, forming together with IRF9 the trimeric ISGF3 complex. ISGF3 translocates to the nucleus, binding to ISRE and inducing the transcription of numerous ISGs. Apart from the canonical JAK-STAT signaling pathway, other non-classical signaling cascades downstream of the IFNAR are also activated upon IFN binding. Created with BioRender.com.

**Figure 3 viruses-16-00364-f003:**
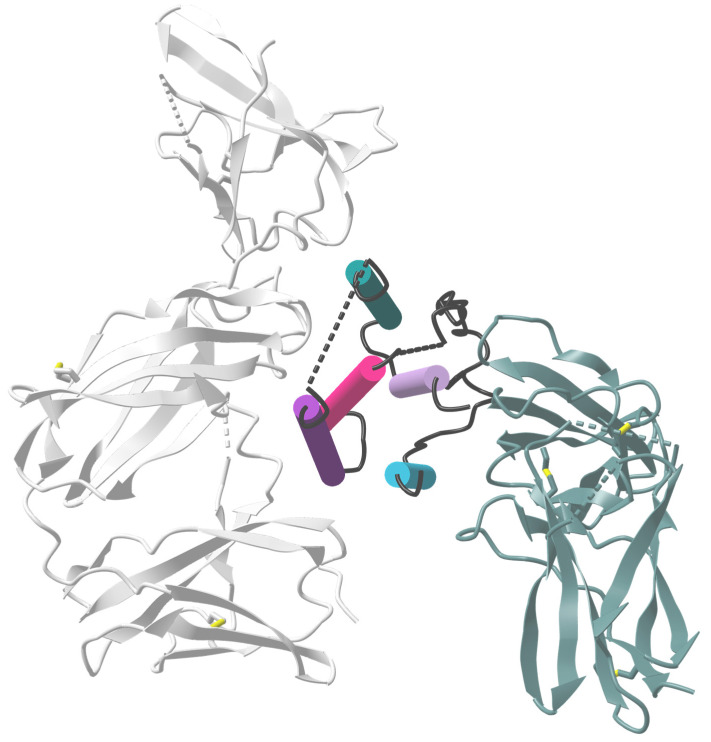
Crystal structure of the IFNAR2-IFNα2-IFNAR1 ternary complex. The ternary IFNAR2-IFNα2b-IFNAR1 complex is depicted with ribbon structures for the receptors and cylinder/plate structure for IFNα2. The five helices of IFNα2 are represented in different colors: aquamarine blue (helix A), pink (helix B), dark purple (helix C), mineral green (helix D), and light purple (helix E). Initially, IFNα binds with a higher affinity to IFNAR2 (shown in dark gray), where helices A and E, along with the AB loop, interact with the D1 and D2 domains of IFNAR2. Subsequently, IFNAR1 (depicted in light gray) is recruited, and its SD1-SD3 domains interact with the helices B, C, and D of IFNα2. This illustration is adapted from [77] and was created using VAST+, PDB ID: 3SE3.

**Table 1 viruses-16-00364-t001:** *IFNA* subtype expression during HIV infection.

Cell Type	HIV Infection/ Exposure	Upregulated *IFNA* mRNAs	Detection Method	References
Isolated pDCs	Patients with HIV (CDC stage A or C)	*IFNA6* and *IFNA2*	RT-PCR	[35]
	Patients with HIV (CDC stage C)	*IFNA1/13*, *IFNA8*, *IFNA14*, *IFNA16*, *IFNA17*, and *IFNA21*		
PBMCs	ART-treated chronically HIV-positive patients	*IFNA2*, *IFNA4*, *IFNA5*, *IFNA6*, *IFNA7*, *IFNA14*, and *IFNA16*	RT-PCR	[36]
	ART-naïve chronically HIV-positive patients			
PBMCs	ART-naïve chronically HIV-positive patients	*IFNA1/13*, *IFNA2*, *IFNA5*, and *IFNA4*	Illumina sequencing	[38]
Isolated pDCs	Exposed to HIV-1_BaL_	*IFNA2* and *IFNA14*	Illumina sequencing	[37]

## Data Availability

This review article does not contain new data to make available.

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
