# Peer review of "IFNα Subtypes in HIV Infection and Immunity"

_viruses, 2024, doi:10.3390/v16030364_

Round 1

Reviewer 1 Report

Comments and Suggestions for Authors

In this review, describe the induction and function of different subtypes of interferon alpha in the context of HIV/SIV infection. This review highlights the literature available on the roles of different subtypes of interferon alpha during HIV/SIV infection. 

Minor Comments: 

Please introduce the 13 different subtypes of interferon-a

Line 16 says 12 interferon-a subtypes and line 46 says 13 different IFN-a genes

Line 78: “induction of IFNs nicely correlated with plasma viral loads” is this a positive or negative correlation. Is it the timing of interferon induction? 

Line 78-80: Please mention the sample or cells used for determining the expression of ifn-a subtype mRNA

Do infected CD4 T cells express different subtypes of IFN-a mRNA?

Line100-103: again, the correlation with viral loads is not clear. 

Line 108: Does mass spec analysis identify different subtypes of IFN-a

Figure 1 legend mentions PRRs Trex1 and SAMHD1, but they are not depicted on the figure

Line 250. Please correct the sentence

Line 252-254: please correct the sentences. They are not conveying the correct meaning. 

Line 270: Should be SIV viremia

Line 271-272 is not very clear

How about some of the HIV-1 or SHIV adapting to type 1 interferon response?

Author Response

Minor Comments: 

Reviewer 1 (R1): Please introduce the 13 different subtypes of interferon-a

Karakoese et al (K): See next comment.

R1: Line 16 says 12 interferon-a subtypes and line 46 says 13 different IFN-a genes

K: The reviewer is correct in noting that there are 13 different IFNA genes; however, it is important to highlight that these genes encode for only 12 distinct proteins, since the mature proteins of IFNα1 and IFNa13 are identical. We addressed this discrepancy by mentioning all gene names in the text and providing further explanation in the manuscript (lines 44-46).

R1: Line 78: “induction of IFNs nicely correlated with plasma viral loads” is this a positive or negative correlation. Is it the timing of interferon induction? 

K: There is a positive correlation observed, in which increasing viral titers correspond to higher levels of detectable IFNα in the serum. We have adjusted the text to reflect this finding accordingly.

R1: Line 78-80: Please mention the sample or cells used for determining the expression of ifn-a subtype mRNA

K: We added the missing information to line 86.

R1: Do infected CD4 T cells express different subtypes of IFN-a mRNA?

K: Unfortunately, we are unable to provide a response to this question. Previous studies have demonstrated that the majority of type I IFN response in HIV infection is generated by plasmacytoid dendritic cells (pDCs). PBMCs depleted of pDCs exhibited a significant decrease in IFN production compared to total PBMCs, while purified pDCs were found to produce substantial amounts of IFN (Lepelley et al., 2011; https://doi.org/10.1371/journal.ppat.1001284). However, it is important to note that every cell has the capability to produce IFN, and it cannot be ruled out that the pattern of IFN expression in an infected T cell may differ from that of an uninfected T cell.

R1: Line100-103: again, the correlation with viral loads is not clear. 

K: We apologize for the oversight in the manuscript and have made the necessary changes to address the missing information.

R1: Line 108: Does mass spec analysis identify different subtypes of IFN-a

K: The reviewer's suggestion is correct. In fact, mass spectrometry analysis could indeed be utilized to distinguish between the different IFNα subtypes at the protein level. However, it is important to note that the assays should be adapted to accommodate various biological samples, such as serum, plasma, cell culture supernatants. Additionally, the assays should be capable of high-throughput analysis and detecting low concentrations of the analytes. We have included the mention of mass spectrometry analysis in the text (line 121).

R1: Figure 1 legend mentions PRRs Trex1 and SAMHD1, but they are not depicted on the figure.

K: We deleted both terms from the figure legend.

R1: Line 250. Please correct the sentence

K: We apologize for this error and corrected the sentence accordingly.

R1: Line 252-254: please correct the sentences. They are not conveying the correct meaning. 

K: We apologize for the unclear description and have revised the sentence accordingly.

R1: Line 270: Should be SIV viremia

K: We changed the text accordingly.

R1: Line 271-272 is not very clear

K: We apologize for the unclear description and have changed the sentence accordingly.

R1: How about some of the HIV-1 or SHIV adapting to type 1 interferon response?

K: The reviewer raised an interesting point regarding HIV or SIV's ability to antagonize the IFN response through various accessory proteins such as Vpu or Vif, which target tetherin or APOBEG3G. However, to our knowledge, there are currently no data available on HIV/SIV evasion strategies specifically targeting the different IFNα subtypes. Therefore, we have opted not to include discussions on HIV/SIV's potential to evade the IFN response in this review.  

Reviewer 2 Report

Comments and Suggestions for Authors

General comments 

-       The topic is relevant and of interest for a broad audience.

-       The review covers the main chapters of the IFN response.

-       Authors are knowledgeable and have previously written reviews on this topic.

-       The present review covers some aspects that were not developed in their previous reports, notably in part 3 and 4.

Main points

-       The main added value of this review is the update of work done in the last few years, but the authors cite almost exclusively their work after 2020.

-       Several statements are too superficial e.g.: Line 42 how do IFN link innate and adaptive immunity? line 64 how do PRR contribute to HIV restriction?

-       Lines 59-76, the description goes back and forth, it would be best if these events were enumerated sequentially (a, b, c…).

-       Lines 60-63, a paragraph describing how PRR sense replication intermediates and how they trigger IFN production is necessary. 

-       Previous work performed in part by the authors and reported on page 2 lines 77 to 95 is hard to follow, as different subtypes are induced in different systems. The authors could help making sense of the complex findings. 

-       Similarly, the description of the paragraph from line 337 to 348 (page 10) is fragmented, and intermediate conclusions would facilitate the understanding. 

-Line 230 - 243, an illustration of the structure of the IFN molecule is necessary.

Minor points,

The binding to the receptor is described twice, the presentation in the introduction should be deleted.

Several references are not formatted.

“alpha” and “beta” are A and B in part of the text.

Line 88, indicate which are the three (out of 5) subtypes induced in the mock-stimulated condition, to allow identification of the specifically induced ones. 

Line 105 “Despite the importance…” the sentence is not clear and the reference is missing.

Line 163, “Interestingly, …” reword.

Line 201, the sentence “The orchestration … needs a reference.

Line 250 “…treatment had to antiviral effect” – replace TO with NO.

Line 253, “As therapy…” sentence incomplete.

A few other sentences contain minor mistakes. 

Comments on the Quality of English Language

The quality of English is good, only a few sentences have been written too fast.

Author Response

Reviewer 2 (R2):

-       The topic is relevant and of interest for a broad audience.

-       The review covers the main chapters of the IFN response.

-       Authors are knowledgeable and have previously written reviews on this topic.

-       The present review covers some aspects that were not developed in their previous reports, notably in part 3 and 4.

Karakoese et al. (K):  We thank the reviewer for the overall positive comments.

Main points

R2:-       The main added value of this review is the update of work done in the last few years, but the authors cite almost exclusively their work after 2020.

K: We apologize if it seemed that we only referenced our own work. Our intention was to provide a comprehensive overview of IFNα subtypes in retroviral infections, incorporating new findings since our previous review in 2018. It is worth noting that the number of researchers focusing specifically on IFNα subtypes (as opposed to just IFNα and IFNβ) in HIV/SIV infections worldwide is quite small. As a result, some of our own work has been cited here. However, we also included a significant number of studies from other researchers, including those utilizing humanized mice or SIV studies that were not conducted by us. Additionally, we have now incorporated more studies from other groups, such as Tauzin et al., 2021; Swainson et al., 2022; and Carnathan et al., 2018.

R2:-       Several statements are too superficial e.g.: Line 42 how do IFN link innate and adaptive immunity? line 64 how do PRR contribute to HIV restriction?

K: We apologize for the superficial statements within the text. We carefully reviewed the manuscript and revised the text accordingly. Given the constraints of the introduction, a more comprehensive elucidation can be found in the chapter titled 'Modulation of Immune Cell Functions by IFNα Subtypes during Retroviral Infections.'

R2:-       Lines 59-76, the description goes back and forth, it would be best if these events were enumerated sequentially (a, b, c…).

K: We concur with the reviewer's feedback and have thoroughly revised and reorganized this paragraph.

R2:-       Lines 60-63, a paragraph describing how PRR sense replication intermediates and how they trigger IFN production is necessary. 

K: We have integrated additional information regarding the downstream signaling cascades of various PRRs and the induction of type I IFN into this paragraph (lines 62-67).

R2:-       Previous work performed in part by the authors and reported on page 2 lines 77 to 95 is hard to follow, as different subtypes are induced in different systems. The authors could help making sense of the complex findings. 

K: To clarify the data described in the paragraph on lines 87-106, we have added a table summarizing the main findings of IFNA subtype gene expression from various studies.

R2:-       Similarly, the description of the paragraph from line 337 to 348 (page 10) is fragmented, and intermediate conclusions would facilitate the understanding. 

K: We apologize for the fragmented content of this paragraph. To enhance clarity, we have added a conclusion regarding the diverse immunomodulatory actions of IFNs in humanized mouse models (lines 364-366).

R2:-Line 230 - 243, an illustration of the structure of the IFN molecule is necessary.

K: We concur with the reviewer's assessment that illustrating the structure of IFNα in complex with both receptor subunits is crucial to improve the understanding of the paragraph. As such, we included the attached figure as Fig. 3 in the manuscript:

Minor points,

R2:The binding to the receptor is described twice, the presentation in the introduction should be deleted.

K: We apologize for the repeated description of the receptor binding. We have deleted a detailed description from the introduction part.

R2:Several references are not formatted.

K: We have formatted the references accordingly.

R2: “alpha” and “beta” are A and B in part of the text.

K: Within the text, we have distinguished between IFN genes names as IFNA or IFNB and IFN proteins labeled as IFNα or IFNβ.

R2:Line 88, indicate which are the three (out of 5) subtypes induced in the mock-stimulated condition, to allow identification of the specifically induced ones. 

K: We acknowledge the mentioned point and have subsequently adjusted the paragraph accordingly (line 97).

R2:Line 105 “Despite the importance…” the sentence is not clear and the reference is missing.

K: We have revised the sentence accordingly.

R2:Line 163, “Interestingly, …” reword.

K: We have revised the wording accordingly.

R2:Line 201, the sentence “The orchestration … needs a reference.

K: We have added an appropriate reference (McNab et al, 2015).

R2:Line 250 “…treatment had to antiviral effect” – replace TO with NO. 

K: We have revised the wording accordingly.

R2:Line 253, “As therapy…” sentence incomplete.

K: We have revised the sentence accordingly (line 272).

R2:A few other sentences contain minor mistakes. 

K: We apologize for any spelling and grammatical errors. We carefully reviewed through the entire manuscript and made revisions as necessary.
